# Leveraging Large Language Models to Estimate Clinically Relevant Psychological Constructs in Psychotherapy Transcripts

**RESEARCH ARTICLE –
SPECIAL ISSUE**

]u[ ubiquity press

**MOSTAFA ABDOU** ⓘ

**RAZIA S. SAHI** ⓘ

**THOMAS D. HULL** ⓘ

**ERIK C. NOOK\*\*** ⓘ

**NATHANIEL D. DAW\*\*** ⓘ

*Author affiliations can be found in the back matter of this article
\*\*These authors contributed equally to this work as last authors

## ABSTRACT

Developing precise, innocuous markers of psychopathology and the processes that foster effective treatment would greatly advance the field's ability to detect and intervene on psychopathology. However, a central challenge in this area is that both assessment and treatment are conducted primarily in natural language, a medium that makes quantitative measurement difficult. Although recent advances have been made, much existing research in this area has been limited by reliance on previous-generation psycholinguistic tools. Here we build on previous work that identified a linguistic measure of "psychological distancing" (that is, viewing a negative situation as separated from oneself) in client language, which was associated with improved emotion regulation in laboratory settings and treatment progress in real-world therapeutic transcripts (Nook et al., 2017, 2022). However, this formulation was based on context-insensitive word count-based measures of distancing (pronoun person and verb tense), which limits the ability to detect more abstract expressions of psychological distance, such as counterfactual or conditional statements. This approach also leaves open many questions about how therapists' — likely subtler — language can effectively guide clients toward increased psychological distance. We address these gaps by introducing the use of appropriately prompted large language models (LLMs) to measure linguistic distance, and we compare these results to those obtained using traditional word-counting techniques. Our results show that LLMs offer a more nuanced and context-sensitive approach to assessing language, significantly enhancing our ability to model the relations between linguistic distance and symptoms. Moreover, this approach enables us to expand the scope of analysis beyond client language to shed insight into how therapists' language relates to client outcomes. Specifically, the LLM was able to detect ways in which a therapist's language encouraged a client to adopt distanced perspectives—rather than simply detecting the therapist themselves being distanced. This measure also reliably tracked the severity of patient symptoms, highlighting the potential of LLM-powered linguistic analysis to deepen our understanding of therapeutic processes.

**CORRESPONDING AUTHOR:**
**Mostafa Abdou**

Princeton Neuroscience Institute, Princeton University, US

ma4231@princeton.edu

**KEYWORDS:**
computational modelling; depression; anxiety; language models; linguistic distancing

**TO CITE THIS ARTICLE:**
Abdou, M., Sahi, R. S., Hull, T. D., Nook, E. C., & Daw, N. D. (2025). Leveraging Large Language Models to Estimate Clinically Relevant Psychological Constructs in Psychotherapy Transcripts. *Computational Psychiatry* 9(1): pp. 187–209. DOI: https://doi.org/10.5334/cpsy.141

Abdou et al.
*Computational Psychiatry*

**188**

# 1 INTRODUCTION

Language is the primary medium through which people think and communicate about their daily subjective and emotional experiences (Rimé, 2009; Brans et al., 2014). But despite a long history of work that recognizes the salience of language as a window into the human psyche (Freud, 1966; Allport and Vernon, 1930; Allport, 1942; Laffal, 1964; Gottschalk and Gleser, 2022; Pennebaker and Stone, 2003; Tausczik and Pennebaker, 2010; Jackson et al., 2022), the promises of linguistic analysis for the psychological and psychiatric sciences have remained in part unfulfilled due to difficulties in delimiting and quantifying the wide range of complex patterns that occur in naturalistic linguistic data.

Indeed, a majority of previous work in mental health research has used earlier-generation natural language processing methods, particularly those based on word counts and dictionaries (such as Linguistic Inquiry and Word Count, LIWC; Pennebaker 2001). These tools, which count words belonging to laboriously compiled lists of psychologically suggestive categories, offered researchers the possibility of linking spontaneous language use to a wide range of behavioral data and have led to crucial insights (Tausczik and Pennebaker, 2010; Newman et al., 2003; Kahn et al., 2007; Holmes et al., 2007; Leshed et al., 2007; Newman et al., 2008; Kacewicz et al., 2014). However, reliance on word frequency alone as a primary measure of psychological constructs also leads to key limitations. Most notably, such counts do not capture the structure and context-sensitivity essential to natural human language (Eichstaedt et al., 2021).

Recent developments in language technology, however, have opened up the possibility of analyses that can quantify meaning more sensitively and address those shortcomings. Such methods (known as Large Language Models; LLMs) encode language into vector embeddings that depend not just on word identity but on surrounding context, capturing a wide array of morphological, syntactic, semantic, and discursive features. General-purpose "foundation models" (i.e., LLMs pre-trained across extensive, varied language corpora and tuned for instruction-following, thereby able to adapt to a range of tasks via in-context prompting) have revolutionized a range of natural language applications from language comprehension and generation to machine translation (Brown et al., 2020; Achiam et al., 2023; Touvron et al., 2023) and has similar promise for psychopathology. Importantly, these models not only capture linguistic meaning more accurately than word-counting approaches, but they can also be leveraged to bring their massive data-driven pre-training to new tasks (such as assessing novel psychological constructs) by straightforward instruction without designing bespoke word lists or classifiers.

Here, we contribute to an emerging body of research that leverages these methods for psychological and psychiatric research—for example, to detect suicidal ideation in social media posts, generate interpretable symptom-level predictions from online text, and identify symptom-relevant content and summaries from clinical interviews (Low et al., 2020; Burkhardt et al., 2022; Malgaroli et al., 2023; Demszky et al., 2023; Zuromski et al., 2024; Chim et al., 2024a; Wang et al., 2024; Yang et al., 2024; Uluslu et al., 2024; Sharma et al., 2024; Mangalik et al., 2024; Chim et al., 2024b; Blanco-Cuaresma, 2024; Jeon et al., 2024; Rathje et al., 2024; So et al., 2024). We apply these models to extend recent research on a theory-based linguistic construct that has thus far only been formulated using word-count approaches. Specifically, previous work has demonstrated that linguistic strategies that create psychological distance — such as shifting from first-person pronouns ("I") to second- or third-person pronouns, using past or future tense rather than present tense, and minimizing self-referential language — are linked to improved emotional regulation (Nook et al., 2017, 2020; Kross et al., 2014, 2017; Orvell et al., 2021). These distancing techniques are thought to help individuals adopt a perspective with a higher level of construal (Trope and Liberman, 2010; Moran and Eyal, 2022; Dercon et al., 2024), reducing the intensity of negative feelings and facilitating self-regulation. Termed "linguistic distancing," this phenomenon has been found to play a role in psychological resilience. For example, in a dataset of more than 6,000 real-world psychotherapy transcripts, linguistic distancing predicts internalizing symptom severity and tracks treatment progress (Nook et al., 2022), with similar results in adolescent single-session interventions (Cohen et al., 2022).

Abdou et al.   **189**
*Computational Psychiatry*

Here, we build on this work by leveraging LLMs' more sophisticated language understanding capabilities to measure linguistic distancing. The context sensitivity of these models allows them to much more nimbly assess meaning, tone, and subtle nuances in language; and their instructability allows us to prompt them to do so by simply providing a definition of the desired construct in language largely drawn from the academic literature. This approach may enable more comprehensive and precise linguistic measurement of the underlying construct of psychological distance, potentially strengthening our estimates of client mental health from language. Furthermore, the flexibility of LLMs also allows us to extend the analysis beyond client language to examine how *therapists*' language is associated with therapeutic outcomes.

To date, it remains unknown whether therapist linguistic distance is related to treatment outcomes, even though establishing such a relationship could identify a simple linguistic strategy for boosting treatment outcomes. However, there are several ways in which therapeutic distancing could relate to outcomes. First, therapists who are using more psychologically distanced language may (i) themselves be better regulated in-the-moment or (ii) be modeling to their client how to maintain a distanced and regulated perspective, both of which could help clients better manage their emotions (Nook et al., 2023). Either of these pathways (assessed using LIWC or LLM metrics) would support a relationship between higher therapist linguistic distance and better client symptom changes. By contrast, an even more abstract process could occur within the dyadic setting of psychotherapy in which the therapist explicitly *encourages* their client to take a distanced perspective on their problems as a form of interpersonal emotion regulation (Zaki and Williams, 2013; Sahi et al., 2021, 2023). This could mean directly providing new perspectives on clients' experiences or asking questions that prompt the clients themselves to reorient their perspectives. Indeed, research suggests that people often seek emotional support by sharing experiences to receive either socio-affective support (comfort and validation), which is thought to be helpful for short-term relief, or cognitive support (help with reappraisal of negative experiences), which is thought to help in terms of long-term outcomes (Rimé, 2009; Nils and Rimé, 2012; Brans et al., 2014). Moreover, Cognitive Behavioral Therapists are specifically trained to facilitate "cognitive restructuring" as an important technique to help patients identify and alter unhelpful patterns of thought (Beck, 2020). These types of abstract strategies employed by therapists are unlikely to be manifest through simple word-counting techniques, marking a clear opportunity for LLM-based approaches to demonstrate their advantage.

Our results demonstrate an advantage for appropriately-prompted LLMs over traditional methods in relating client language to clinical outcomes, clarifying the shortcomings of traditional NLP for abstract judgments of this sort; and, extending for the first time in this line of work to therapist language, demonstrating that therapist language that encourages clients to adopt a distanced perspective (distinct from using distanced language themselves) also correlates with clients' symptom trajectories. This last finding suggests that LLM-based linguistic analysis can offer novel insights into how therapists' communication styles and use of interpersonal regulatory strategies shape the therapeutic process and contribute to symptom improvement. Overall, we highlight the potential of LLM-powered linguistic analysis as a tool for advancing research in clinical psychology and refining therapeutic techniques.

## 2 MATERIALS AND METHODS

This study builds directly on Nook et al. (2022), using the same dataset and analytical choices wherever possible but comparing novel LLM-based measures to previous word-based methods for quantifying text. Accordingly, we summarize central methods here but refer readers to Nook et al. (2022) for a more detailed description of the dataset and previous methodological approach.

### 2.1 DATASET

This study analyzed data from a random sample of 3,727 clients who used the digital therapy service Talkspace from 2016 to 2019.[1] Following Nook et al. (2022), participants were included

---

1   Talkspace.com.

only if they completed at least three symptom assessments over a minimum span of six weeks. After applying these selection criteria, Nook et al. (2022) divided the remaining pool of clients (6, 229) into exploratory (3, 727) and validation sets (2, 500). Only the former is utilized in this study. Talkspace is a digital mental health platform that provides session-based teletherapy as well as asynchronous text-message-based therapy. Most patients in the study opted for the latter, and this is the source of the transcripts we analyze. The data analyzed previously included 455, 379 messages; we extend the analysis to include 273, 208 messages from 1, 359 therapists to their clients, not considered in the original study. Note that we only analyze the exploratory subset of the data used by Nook et al. (2022). This is because the new study involved minimal analytic flexibility since it used the earlier study's analyses and code where possible, substituting only revised explanatory variables. Accordingly, we chose to reserve the validation dataset from the earlier study (messages from an additional 2, 500 clients used to replicate the original findings) for future analyses.

Participants also completed at least three symptom questionnaires over a minimum of six weeks of treatment. For each client, researchers accessed (1) anonymized records of text exchanges with their therapist, (2) their responses to depression and anxiety assessments, and (3) demographic details.

Consent for research use was granted under Talkspace's terms of service. The Princeton University IRB determined that study plans were not human subjects research. Due to the sensitive nature of the data and the requirements of the data-use agreement, we housed both data and models in a secure, local setting rather than using cloud storage or compute services.

## 2.2 SYMPTOM ASSESSMENT

Clients received symptom questionnaires roughly every three weeks via the Talkspace messaging platform. Completion dates for each questionnaire were recorded and converted to represent the participant's time in therapy based on the number of days that had passed since the first text with their primary therapist. Depression symptoms were assessed using the PHQ-8 (Kroenke et al., 2009), a standard tool for measuring depression severity, which covers eight of the DSM-4's nine defining symptoms of major depressive disorder: anhedonia, low mood, sleep disturbance, fatigue, appetite disturbance, low self-esteem, concentration difficulties, and psychomotor agitation or slowing. (The widely used PHQ-9 additionally includes suicidal ideation, but the reduced scale omitting this item is validated with similar sensitivity and reliability; Razykov et al. 2012; Shin et al. 2019.) Responses were made on a 0–3 scale, summed to an overall measure between 0 and 24. Anxiety symptoms were measured using the GAD-7, which assesses seven indicators of generalized anxiety disorder (feelings of anxiety, uncontrollable worry, excessive worry, difficulty relaxing, restlessness, irritability, and fears of catastrophic outcomes). Responses were made on a scale from 0–3 each, summed to obtain an overall score between 0 and 21. Following Nook et al. (2022), we summed the two measures to produce a single assessment of internalizing symptoms, as PHQ-8 and GAD-7 measures were highly correlated ($r = 0.757$, $p < 0.0001$). This strong relationship aligns with prior work showing that depression and anxiety commonly co-occur and load onto a shared internalizing factor in dimensional models of psychopathology (Gillan et al., 2016; Newman, 2022; Wise et al., 2023).

## 2.3 LARGE LANGUAGE MODELS

We utilize the open source LLaMA (Large Language Model Meta AI) version 3.1 models (AI@Meta, 2024), a generative class of models which has exhibited high performance on a wide range of linguistic tasks (Dubey et al., 2024). Additionally, these models are publicly available at https://www.llama.com/llama3_1/. For all main experiments, we use the 70-billion parameter version of the model loaded via the `Hugging Face` python library,[2] further quantized to 4 bits of precision per parameter. Quantization is carried out using methods from the `bitsandbytes` library,[3] where

---

2      https://huggingface.co/meta-llama/Llama-3.1-70B-Instruct.

3      https://github.com/bitsandbytes-foundation/bitsandbytes.

nested quantization is applied to reduce the size of each model weight from float32 (32 bits, the original precision) to bfloat16 (16 bits). We chose this model size and quantization because it fits on a single NVIDIA *A*100 GPU, rendering inference on a large dataset feasible. We also report comparisons to the same analyses run on the smaller 8-billion parameter model with identical quantization settings in Appendix A.1.[4] We do not train or fine-tune the models' parameters on the message data, but instead use their out-of-the-box inference capacities via prompting. This circumvents possible data privacy concerns, as text data are never sent from one's machine to another device or remote server.

## 2.4 LINGUISTIC DISTANCING

Linguistic distancing was assessed, message by message, using three methods across the experiments. The goal of each method was to map each message to a numeric score; these were then averaged over all the messages in the (approximately 3 weeks) period between each symptom assessment and used as variables in regression analyses (below).

The first method (LIWC self-distance) corresponds to Nook et al. (2022)'s LIWC-based approach, which calculates two key metrics (temporal distance and social distance) which are then averaged to produce a single composite measure of linguistic distance. Temporal distance was measured as the proportion of verbs that were not in the present tense, using the formula (past+future)/(past+future+present). Social distance was calculated as the proportion of pronouns that were not first-person singular, using (second person+first-person plural+third-person singular+third-person plural)/(total pronouns). Verb and pronoun counts were computed programmatically using LIWC software.[5] Temporal distance scores were impossible to compute for messages without verbs (7.1% of the data), and social distance scores were impossible to compute for messages without pronouns (9.4% of the data). To enable direct comparison across methods on the same dataset, these messages are excluded from all analyses, even though the LLM-based method is able to compute a measure of distancing for them.

The second method (LLM self-distance) employs LLMs to assess psychological distance. A linguistic prompt was input to the model along with a given textual datapoint (written by either a client or therapist). The LLM is prompted with an explanation of linguistic distancing and instructed to score the message on this construct by outputting a measure between 1 and 5 (see subsection 2.5 for details).

The final method (LLM other-distance, used with therapist text messages only) also employs LLMs to assess distance; however, rather than assessing how distanced from the self a text is, the prompt instructs the model to assess the extent to which the therapist encourages the client to adopt a distanced perspective.

Distancing scores were averaged across messages without weighting by length, following Nook et al. (2022). While this gives equal weight to short and long messages, we retained the prior study's approach for comparability and to avoid overemphasizing longer messages that may not be more informative. Nevertheless, we note that message length (measured as a word count) does show some correlation with LLM self-distance ($r = 0.16$; $p < .001$), but the significance of our results does not change when controlling for word count.

## 2.5 MODEL PROMPTING

The prompts used to derive LLM self-distance and LLM other-distance are shown below. We used a detailed prompt to prioritize conceptual clarity and ensure task alignment. While excessively long prompts can impair model performance, prompt-engineering research shows that well-structured, moderately long prompts often enhance LLM performance—particularly for complex, domain-specific tasks (Liu et al., 2025; Long et al., 2025).

---

4     https://huggingface.co/meta-llama/Llama-3.1-8B-Instruct.

5     https://www.liwc.app/.

Abdou et al.
*Computational Psychiatry*

**192**

For LLM self-distance, the prompt (seen in Figure 1) defines psychological distance and describes how it might manifest linguistically. A scale with five levels (*A-E*) is then described and the model is instructed to output one of those labels to quantify the degree of linguistic distancing in a given text. This measure of linguistic distance is applied to both client and therapist messages.

---

**LLM self-distance Prompt**

Below, we ask you to rate a passage of text according to how the language used reflects psychological distance. To explain the concept, consider the following:

People are capable of thinking about the future, the past, remote locations, another person's perspective, and counterfactual alternatives. These constitute different forms of traversing psychological distance. Psychological distance is egocentric: its reference point is the self in the here and now, and the various ways in which an object might be removed from that point—in time, in space, in social distance, and in hypotheticality—constitute different distance dimensions.

Furthermore, people can use language to psychologically distance themselves from a particular situation (i.e., "taking a step back" and seeing the situation as separated from themselves). We refer to this process as linguistic distancing. One possible way to increase linguistic distance is to speak in the second or third person (e.g. 'you' or 'they') rather than in the first person ('I'). Another is to talk in the past or future tense instead of the present tense. These are two examples, but people can also express linguistic distance in other ways, including speaking of distant places or hypotheticals.

For a given text, please rank how much the speaker uses some form of linguistic distancing: that is, how separate or distant the text is from the speaker's self. To do so, choose one of the following options, which range from A to E:

(A) very low distance

(B) low distance

(C) medium distance

(D) high distance

(E) very high distance

Your vocabulary consists only of the following: A, B, C, D, E.
Note that '[-]' indicates that a proper name of a person or place was removed for privacy.
Never output any other characters.

Text: **\*Text to be rated inserted here\***
Rank:

---

**Figure 1** Textual prompt used to obtain ratings of self-distance.

For LLM other-distance (seen in Figure 2), the prompt indicates that the message is written by a psychotherapist to a client. It similarly explains psychological distancing, but instead of describing how distancing may be reflected in language, it describes ways in which a therapist might help a client to adopt a distanced perspective. Finally, it asks the model to evaluate to what extent the language used encourages the client to adopt a distanced perspective. This measure of linguistic distance is applied to therapist messages only.

We prepended these instructions to each message—truncated after 1000 tokens due to empirically determined GPU memory limits—and input it to the network to extract a score.[6] In standard use, these networks are trained to generate text stochastically, sampling each token from a next-word probability distribution conditional on the preceding text. To compute a reproducible score from a single run, we extract the model's distribution over the five permissible tokens and, rather than sampling from it, compute the expected output on average over the rankings.

---

6    Only the message text was truncated after 1,000 tokens, meaning that all text that is part of a message that exceeded the first 1,000 tokens was discarded. The instruction prompt before and after the message was kept fully intact. Such truncation occurred in only 0.55% of messages.

Specifically, the model quantifies the predictive distribution by outputting a score in units of logits (an arbitrary real number) for each token. This set is transformed and normalized into a probability distribution (i.e., a set of scores between 0 and 1 that sum to 1, here limited to the five relevant answers out of the model's total vocabulary of 128k tokens), using a logistic softmax function. A higher predicted probability for a particular category label (*A-E*) as the next-word continuation indicates that the model is more confident that the text belongs to that category. These probabilities are then used as weights for computing a weighted average of 1–5, where 1 corresponds to *A*, 2 to *B*, etc. The resulting value, between 1 and 5, quantifies the overall linguistic distance of the text:

$$\text{LLM self-distance} = \sum_{i=1}^{5} i \cdot \text{Softmax}(logit_i)$$

---

**LLM other-distance Prompt**

Below, you will be presented with a text written by a psychotherapist as part of their treatment of a patient during therapy and you will be asked to rate it according to how the language used encourages the patient to employ psychological distancing. To explain psychological distancing, consider the following:

People are capable of thinking about the future, the past, remote locations, another person's perspective, and counterfactual alternatives. These constitute different forms of traversing psychological distance. Psychological distance is egocentric: its reference point is the self in the here and now, and the various ways in which an object might be removed from that point—in time, in space, in social distance, and in hypotheticality—constitute different distance dimensions.

There are several ways in which a therapist can help a patient take a more distanced perspective. For example a therapist might use demonstrations, ask questions, or they might coach or instruct the patient to do so.

For the following text, please rank how much the speaker (the therapist) encourages the patient towards psychological distancing. To do so, choose one of the following options, which range from A to E. The therapist's encouragement of psychological distancing is:

(A)  very low

(B)  low

(C)  medium

(D)  high

(E)  very high

Your vocabulary consists only of the following: A, B, C, D, E.
Note that '[-]' indicates that a proper name of a person or place was removed for privacy.
Never output any other characters.

Text: **\*Text to be be rated inserted here\***
Rank:

---

**Figure 2** Textual prompt used to obtain ratings of other-distance.

## 2.6 STATISTICAL ANALYSES

As our emphasis is on comparing linguistic distancing measures, we closely follow the methods of Nook et al. (2022) for statistical modeling and analyses. We then compare the behavior of LIWC-based and LLM-based measures used as variables in the original models.

Briefly, mixed-effects regressions (using the `lme4` *R* package (Bates et al., 2015) for models and the `lmerTest` package (Kuznetsova et al., 2017) for p-values) were employed to study

how linguistic distance changed over treatment, and how linguistic distance was associated with symptoms. We conducted regressions modeled after those in Nook et al. (2022) but replaced the different measures of distance in each regression. Specifically, we tested two linear relationships: first, regressing days in treatment onto linguistic distance (as measured via the methods described in 2.4) and second, regressing linguistic distance onto internalizing symptoms. For the latter analysis, linguistic distance was decomposed into within-person and between-person components (following Bolger 2013 and Nook et al. 2022). Between-person effects were defined as a participant's average linguistic distance grand mean centered (i.e., their average deviation from the group), while within-person effects are observation scores after participant-centering (i.e., differences between the observation and a participant's mean). When entered together in a regression, these variables allowed us to test whether variance in the outcome variable was associated with person-level variance relative to other clients (between-subjects component) and/or observation-level variance relative to the client's own mean (within-person component). This enabled us to examine both (1) whether an individual's symptoms decreased as their linguistic distance increased (a within-person relationship) and (2) whether individuals with higher overall linguistic distance tended to have lower symptoms compared to those with lower overall linguistic distance (a between-person relationship). Both within- and between-person variables were then simultaneously included in mixed-effects models to examine the relationship between linguistic distance and internalizing symptoms.

All the mixed-effects models included a random intercept for the subject. They also included random slopes for subjects for all explanatory variables that were repeated measures by subject: i.e., for time in treatment when it is an explanatory variable and for within-person linguistic distance when it is an explanatory variable, etc. Linear mixed-effects regression estimates are presented in standardized units ($\beta$), which were calculated using the "pseudo" option of the **standardize_parameters** function from the `effectsize` *R* package (Ben-Shachar et al., 2020), where coefficients are standardized with respect to their "level" (i.e., within-person or between-person variance). Additionally, to assess effect sizes in the mixed-effects models, we report the proportion of variance explained by each independent variable, semipartial $R^2_\beta$, using Satterthwaite's method for estimating degrees of freedom (Edwards et al., 2008; Jaeger et al., 2017).

Next, to formally test whether the candidate measures of linguistic distancing performed statistically differently from one another, we compared the variables' effect sizes between models using a bootstrap. Specifically, we use a paired bootstrap test (comparing two models' performance across datasets resampled over clients or therapists) to test whether the mean semipartial $R^2_\beta$ captured by a predictor variable differed between two measures (e.g., within-person LLM self-distance vs. within-person LIWC self-distance). The procedure for this was as follows. For each of 2,000 re-samplings: (1) $N$ clients or therapists (depending on which analysis it was) were sampled with replacement from the full dataset, where $N$ is the number of clients or therapists in the full data; (2) two mixed-effects regression models were fit on this re-sampled data as described above, one for each of the two measures of linguistic distancing being compared; (3) semipartial $R^2_\beta$ was calculated for the predictor variables corresponding to each of the two measures.

The result of this procedure is an approximate sampling distribution spanned by 2,000 sample pairs of semipartial $R^2_\beta$ scores. Here, each sample represents a simulated replication of the experiment, giving a sample pair of scores (and hence a sample difference between the scores) for two predictor variables, such as within-person LLM self-distance vs. within-person LIWC self-distance. The quantiles of this ensemble define a family of bootstrap confidence intervals over the difference in effect sizes, between methods. We take an effect as significant if the 95% confidence interval (the central 95% of samples) excludes zero.

The result of this procedure is an approximate sampling distribution spanned by 2,000 sample pairs of semipartial $R^2_\beta$ scores. Here, each sample represents a simulated replication of the experiment, giving a sample pair of scores (and hence a sample difference between the scores) for two predictor variables, such as within-person LLM self-distance vs. within-person LIWC self-distance. The quantiles of this ensemble define a family of bootstrap confidence intervals over the difference

Abdou et al.
*Computational Psychiatry*

**195**

in effect sizes, between methods. We take an effect as significant if the 95% confidence interval (the central 95% of samples) excludes zero.

Finally, we also conducted mediation analyses to test whether increases in linguistic distancing mediated reductions in internalizing symptoms. These analyses followed the approach of Nook et al. (2022), utilizing Bayesian regression models. As in the mixed-effects models above, the Bayesian regression models included a random effect for participant. To provide relatively unbiased starting points for Bayesian analyses, we used weakly informative priors (Gaussian distribution of $M = 0$, $SD = 10$). Bayesian analyses were implemented using the Stan language via the `brms` R package (Bürkner, 2017). Two Markov chains utilized the Monte Carlo No U-Turn Sampler to approximate the posterior distribution across 12,500 iterations, with the first 2,500 iterations discarded as burn-in. The indirect effect (i.e., the $a \times b$ pathway for the within-person parameter) and proportion mediated (i.e., indirect effect/[indirect + direct effect] × 100) were computed for each mediation model. A significant mediation was determined when the 95% credible range (CR) of posterior density for the indirect effect excluded zero.

We first replicated Nook et al. (2022)'s original model, replacing LIWC-based distancing with our LLM self-distance measure. Specifically, we tested whether time in treatment predicted increases in within-person distancing, which in turn predicted reductions in symptom measures. We hypothesized that within-person increases in linguistic distance would mediate the effect of time spent in therapy on changes in symptom level, and that this effect would be better captured by LLM self-distance than LIWC self-distance.

Extending Nook et al. (2022)'s work, we also investigated whether therapists' use of distance-encouraging language mediates the effect of time spent in therapy on client symptom levels. Here, we replaced LLM self-distance with LLM other-distance as a mediator. This allowed us to assess whether therapist encouragement of distancing functioned as an interpersonal mechanism of symptom change. In a final setup, we directly model the relationship between LLM other-distance and LLM self-distance, testing how the latter mediates the effect of the former on the level of internalizing symptoms, while controlling for the effect of time spent in therapy.

## 3 RESULTS

### 3.1 CLIENT MESSAGES

We compared our new LLM-based measure of linguistic distancing (LLM self-distance) to the simpler word-count based version used by Nook et al. (2022) (LIWC self-distance). The two measures were moderately correlated across messages (Figure 3; Pearson correlation $r = 0.51; p < .001$). This is consistent with the expectation that they capture related but distinct operationalizations of the same construct, and also confirms that a general-purpose LLM

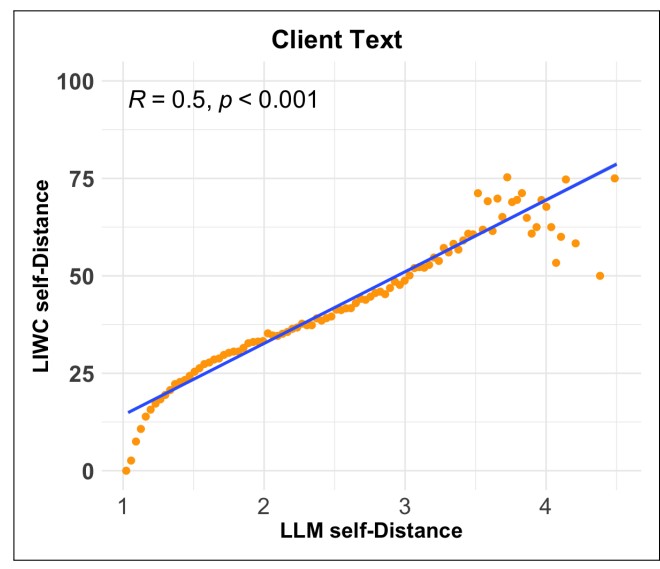

**Figure 3** Correlation between LLM self-distance and LIWC self-distance on 100,000 randomly sampled examples from the dataset. Orange points represent the mean LIWC self-distance within one of 100 bins of the LLM self-distance measure. A linear regression line is overlaid in blue.

prompted with natural language instructions is able to approximate a hand-designed dictionary-based approach.

Abdou et al. **196**
*Computational Psychiatry*

To assess the scientific usefulness and external validity of the new measure, we compared the two measures with respect to the key analyses and results from Nook et al. (2022). Specifically, it was shown there that linguistic distance in client texts increased over time in therapy, and variation in the measure was also associated with decreasing symptoms at both the within-person and between-person levels of analysis (gray bars in Figure 4).

The LLM-based measures reproduced all three effects (gray bars in Figure 4). Specifically, clients' linguistic distancing, as measured by LLMs (LLM self-distance), increased over the duration of treatment (Figure 4): $\beta = 0.09$, $p < 0.001$, $R^2_\beta = 0.004$), and was significantly related to internalizing symptoms, both within-person $\beta = -0.06$, $p < 0.001$, $R^2_\beta = 0.015$ and between-person $\beta = -0.21$, $p < 0.001$, $R^2_\beta = 0.030$.

The effect sizes of LLM-based effects were numerically larger than the LIWC-based ones in all cases, consistent with our hypothesis that the LLM measure would capture more linguistic facets of this abstract construct and thereby exhibit a stronger relationship to internalizing symptoms than the simpler word count based features. Comparing the effect sizes (semipartial $R^2_\beta$) across models using paired bootstraps showed that this improvement was significant for the duration of treatment and for the between-person effect on symptoms but not the within-person effect.

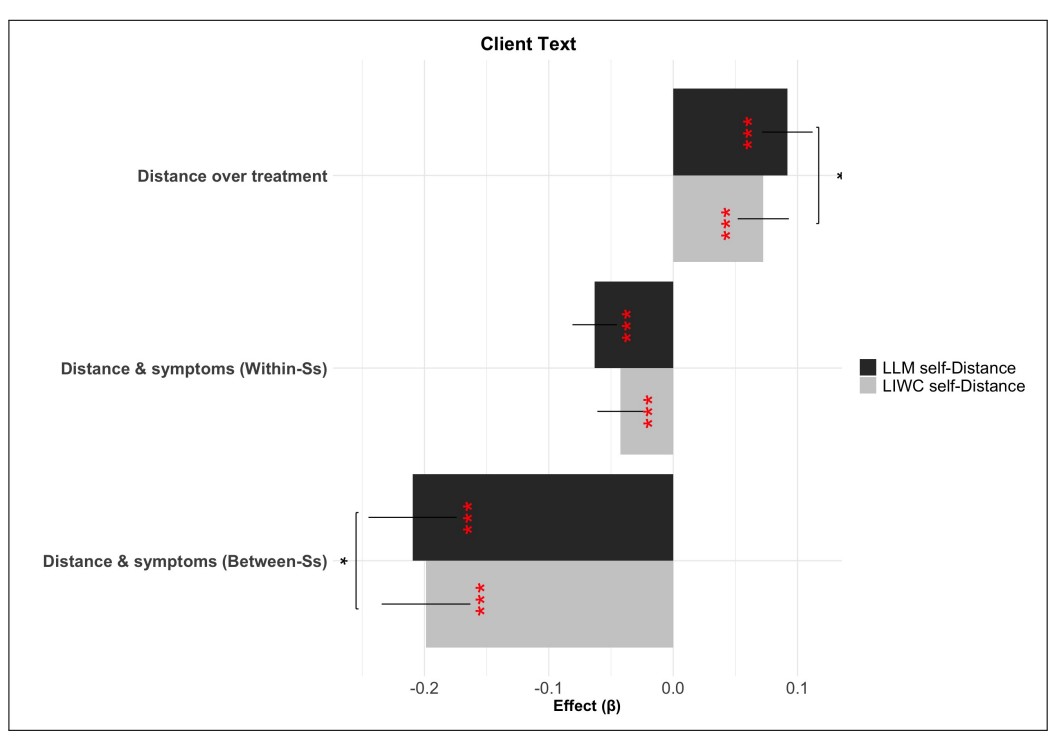

**Figure 4** Effect sizes for mixed-effects regressions, showing relations between internalizing symptoms, linguistic distance, and time in treatment for LIWC self-distance (grey) and LLM self-distance (black) over client text. Effect significance level is indicated in red (i.e., difference between the beta and 0), while significant differences of paired bootstrap comparisons between effects (semipartial $R^2_\beta$) are shown above the relevant pairs of bars in black asterisks. Ss = Subjects.

## Correlations with Other Measures

These results are promising with respect to our hypothesis that LLM-based measures, which can be directly prompted to recognize an abstract construct, might be able to capture and indeed improve upon hand-designed measures. However, the LLM outputs are relatively opaque. Thus, we next undertook a number of analyses aimed at understanding what features of text it might be exploiting, allowing us to characterize its similarities and differences with respect to the count-based features.

As a first step, we computed LLM self-distance's correlation, across messages, with each of the 66 variables from the full set of LIWC features (Figure 10 in Appendix A.2), including the pronoun and tense counts that were aggregated to define LIWC self-distance, plus many other dictionary-based counts. These correlations confirm that the variables with the highest positive

correlations are those associated with second and third-person pronouns as well as past tense verbs (`shehe, social, they, past, we`), and those with the highest negative correlations those of first person pronouns and present tense verbs (`i, present`), together again confirming the viability of natural language prompting to reproduce such measures. Interestingly, a number of other LIWC features show moderate negative correlations with LLM self-distance, including those related to affect, feelings, assent, anxiety, and negative emotion (`affect, feel, assent, and negemo`). These correlations speak to the success of the method in characterizing the underlying construct of psychological distance, which has been associated with reduced negative affect. That said, they may both reflect the LLM taking account of affect in judging distance (e.g., to the extent that preoccupation with affect may be viewed as by definition evidence of lacking a distance perspective), and/or be induced by the dataset itself (i.e., resulting indirectly from the fact that, in this particular corpus, distance, even judged without considering affect, correlates with affective symptoms).

Abdou et al.
*Computational Psychiatry*

## Boundary Samples Analysis

To gain deeper, albeit qualitative, insight into the reasons why the LLM methods outperformed LIWC methods, we next extracted "boundary samples" where LLM self-distance and LIWC self-distance maximally disagreed. We manually reviewed these and drew out themes that captured the qualities that allowed LLM methods to more accurately capture psychological distance than LIWC-based methods in these texts. Due to the sensitive nature of the messages and the requirements of the data-use agreement, we cannot publish the texts themselves. Thus, we rephrased example texts to preserve the style and structure of the original examples while maintaining confidentiality (see Table 1 for summaries). The two distancing measures, shown in the table, were then calculated for each rewritten example, and values were generally consistent with the originals.

The results showcase how simple word counts are a brittle proxy for such an abstract construct, and the boundary messages exemplify cases in which the "spirit" of the language seems to reflect distance (or immersion) even if strict grammatical word usage implies the opposite. Examples of high LLM self-distance and low LIWC self-distance can be found in the top section of the table. Here, we find messages that use first-person pronouns and present-tense verbs, but the language was marked by a high degree of formality, neutrality, introspection, and abstractness. People are reflecting on themselves and others but taking a detached perspective. Although this language is clearly distanced, the LIWC measure of linguistic distance gives it the lowest possible rating because pronouns and verbs are all self-directed and in the present moment.

Conversely, in the bottom half of the table are examples where LLM self-distance is low and LIWC self-distance high. Here, we find language that is informal, emotionally expressive, and grounded in more concrete activities, emphasizing a kind of social and experiential positivity. People are discussing emotionally-arousing and immersive experiences, but they merely drop the first-person pronouns and speak in past-tense. Thus, these examples are not psychologically distanced but are rated as maximally distanced by LIWC self-distance as they do not contain first-person singular pronouns or present tense verbs.

We found similar trends to those observed in Table 1 in the larger pool of original examples. Overall, these examples (though necessarily qualitative) exemplify the ways in which simple grammatical heuristics can fail and showcase the more nuanced and face-valid performance from the LLM.

## Model Size

Because inference with LLMs of this scale requires specialized hardware, we also investigated whether performance was maintained in smaller models of a size roughly feasible for a standard personal workstation. Figure 8 in Appendix A.1 shows mixed effect regression results for the smaller 8-billion parameter model (see section 2.3) on client text, again comparing LLM self-distance and LIWC self-distance. Unlike the larger model used in the main experiments, the smaller model shows inconsistent results with small or non-significant effects. This is consistent

with findings that in many applications, model performance scales with parameter count (Kaplan et al., 2020).

Abdou et al.                    **198**
*Computational Psychiatry*

## 3.2 THERAPIST TEXT

So far, we have shown that a prompted LLM can reproduce and, indeed, slightly improve upon the main results from a previous analysis demonstrating that distance language predicts clinical outcomes. However, these results so far lack a control for specificity and also do not yet showcase the promise of the LLM-based methods to extend beyond the situations where hand-built dictionaries are available to investigate other aspects of language use. For this, we extend the analysis to the messages from clients messages to therapists' messages, which were not considered in Nook et al.'s (2022) original study.

Little is known about how therapists use language to guide patients' adoption of a more distanced perspective. Extending Nook et al.'s (2022) logic, therapists' own use of distanced language might also be associated with clinical improvement: either because it would indicate therapists themselves being more distanced and better regulated during treatment or because it reflects therapists modeling a more distanced perspective for clients. A therapist could, for instance, speak less in the present tense and more in the future or past tense when interacting with their client. A distinct (though not mutually exclusive) possibility is that therapists might instead guide or encourage patients more explicitly about distance-taking strategies without specifically using distanced language themselves. Such guidance could, for instance, be expressed as: "I'd like you to step back and just observe your current problem as if you were in a friend's shoes." (see Table 2 for more examples).To investigate to what extent either of these aspects of therapist language is associated with patients' clinical status, we compared the earlier measure (LLM self-distance), now applied to therapist messages, to a new prompt (LLM other-distance), which asked the model to judge not distanced language usage but instead how the message guided the client in adopting a distance perspective. Any dissociation in results between these prompts would also speak to the specificity of the LLM-measured features.

### Linguistic Distancing Measure Correlations

We observed a Pearson correlation of 0.51 between LLM self-distance and LLM other-distance for therapist messages (Figure 5), a moderate relationship between self-distancing in therapists' language and their encouragement of clients to distance. This relationship suggests that the two constructs, as judged by the LLM, are distinct but related, perhaps because modeling distanced language is one way to promote it.

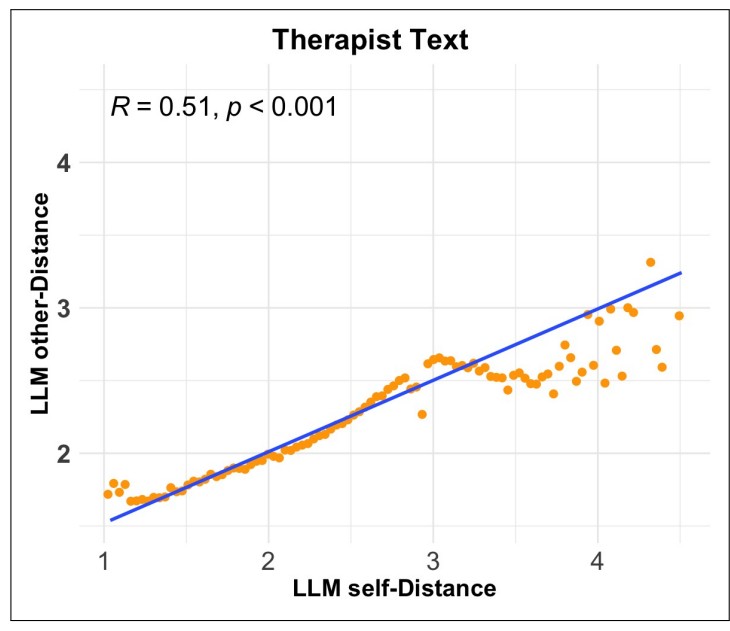

**Figure 5** Correlation between LLM other-distance and LLM self-distance on 100,000 randomly sampled examples from the dataset. Orange points represent the mean LIWC self-distance within one of 100 bins of the LLM self-distance measure. A linear regression line is overlaid in blue.

Abdou et al. **199**
*Computational Psychiatry*

Figure 6 shows results from the same mixed-effect regressions as considered previously, but now predicting patient outcomes on the basis of properties of therapists' language. Like clients' use of distanced language (see Figure 4), variation in therapists' encouragement of distancing (LLM other distance) predicted clients' internalizing symptoms. This relation was significant at both the within- ($\beta = -0.16$, $p < 0.001$, $R^2_\beta = 0.089$) and between-dyad ($\beta = -0.04$, $p < 0.01$, $R^2_\beta = 0.002$) level. In contrast, therapists' own use of distanced language (LLM self-distance) showed no such relationships within- ($\beta = 0.006$, $p = 0.489$, $R^2_\beta = 0.00034$) and between-dyad ($\beta = -0.016$, $p = 0.369$, $R^2_\beta = 0.00021$) level. LLM other-distance for therapist messages, but not self-distance, also increased over the duration of treatment ($\beta = 0.27$, $p < 0.001$, $R^2_\beta = 0.14$), a pattern analogous to patients' progressive adoption of distanced language. Paired bootstraps indicated significant differences between the effect sizes for differences between LLM other-distance and self-distance both in their relations with client symptoms and in their relations with time in treatment.

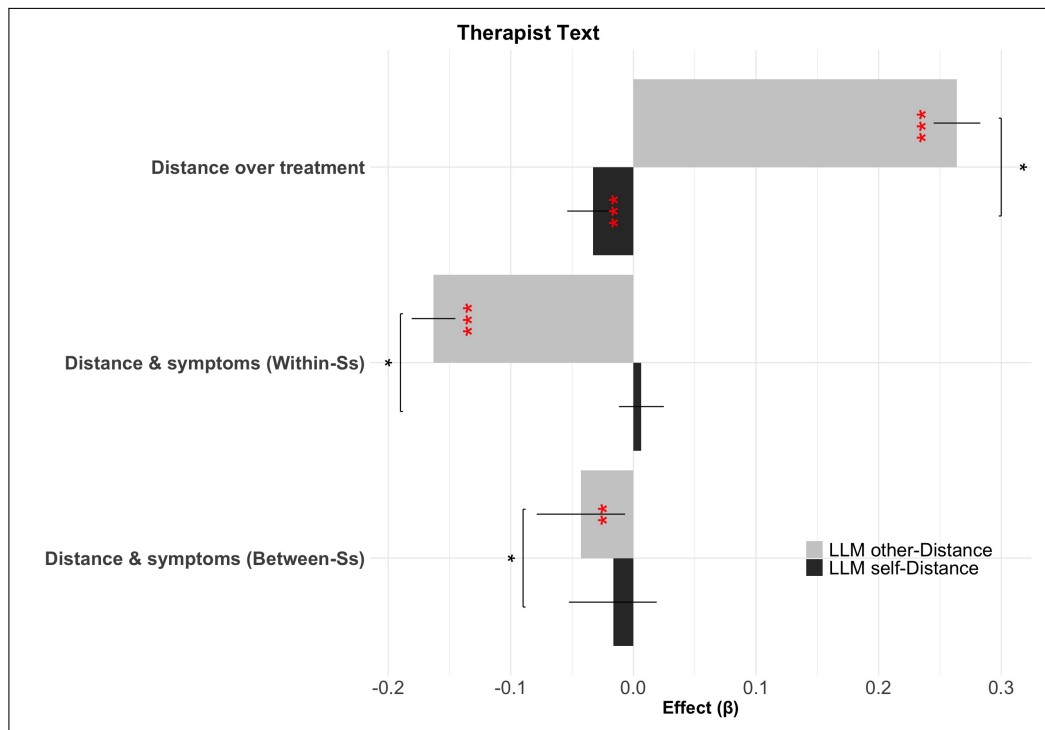

**Figure 6** Effect sizes for mixed-effects regressions, showing relations between internalizing symptoms, linguistic distance, and time in treatment for LLM self-distance (black) and LLM other-distance (grey) over Therapist text. Effect significance level is indicated in red, while results of paired bootstrap comparisons between effects (semipartial $R^2_\beta$) are shown above the relevant pairs of bars. Ss = Subjects.

The finding that LLM self-distance applied to clients' messages, but not therapists', significantly predicts clients' symptoms provides a negative control supporting the specificity of the measure. Meanwhile, the finding that therapists' encouragement of distancing, judged according to LLM other-distance, provides new evidence that an LLM is able to detect clinically relevant language usage in a novel and scientifically relevant setting.

## Boundary Samples Analysis

We next repeated the boundary sample analysis to compare LLM other-distance to LLM self-distance, to investigate what qualitative aspects of therapist language might differentiate distance-promoting language (as judged by the LLM) from distanced language itself. Table 2 shows a set of representative boundary samples where LLM other-distance and LLM self-distance maximally disagree for (again, paraphrased) therapist messages. In the top section of the table are examples with a high LLM other-distance but low LLM self-distance. The language used here is supportive, direct, and instructive, with therapists often using metaphors and other forms of vivid language to help their clients gain perspective. One example (not a direct quote but a fabricated piece of text inspired by therapist content) is "I'd like you to do something for me: step back and just observe your unhappiness. Now, imagine your unhappiness as a wave, coming and going. Picture yourself surfing that wave of unhappiness". For the same reason, the language is often first-person and in the moment: not itself distanced. In the lower section of the table, on the other hand, we find examples where LLM self-distance is high but other-distance (i.e., encouragement of

distance) is low. These are generally examples where the therapists' language is distant and matter of fact, much like the distanced client language in the upper part of Table 1, but not overtly oriented toward promoting the patient to adopt a distanced perspective (e.g., "They seem to have a shared perspective, which both connects them and sometimes leads to tension.", another fabricated example). In general, these examples suggest the LLM is able to detect abstract and meaningful aspects of language use that may be able to drive further discovery.

| LLM SELF-DISTANCE HIGH/LIWC SELF-DISTANCE LOW | | |
|---|---|---|
| **LLM SELF-DISTANCE** | **LIWC SELF-DISTANCE** | **PARAPHRASED EXAMPLE** |
| 7.59 | 0.00 | No matter how the job turns out, it's just a tool, not a goal. It's primarily about finances and being able to afford things, rather than being part of my identity. |
| 7.85 | 0.00 | Every now and then, I feel oddly detached, as though I'm witnessing my own life from a distance, fully aware it's me but feeling as if I'm on the outside looking in. |
| 7.40 | 0.00 | My lawyer and accountant are puzzled by this situation. This process is typically a standard part of any transaction. |
| LLM SELF-DISTANCE LOW/LIWC SELF-DISTANCE HIGH | | |
| **LLM SELF-DISTANCE** | **LIWC SELF-DISTANCE** | **PARAPHRASED EXAMPLE** |
| 1.00 | 10.0 | [-] spent the day at the gym and then joined Mom and her parents for a film shoot! It was such an awesome experience! |
| 1.98 | 10.0 | On Saturday, we slept in, went to yoga, and played cards—it was such a blast! Overall, it turned out to be a fantastic weekend. Haha! |
| 3.10 | 10.0 | We had a fantastic time at a concert tonight! The performances were outstanding, and it was so enjoyable to share the evening together. |

**Table 1** Client text boundary examples where LLM self-distance High and LIWC self-distance Low maximally disagree.

| LLM OTHER-DISTANCE HIGH/LLM SELF-DISTANCE LOW | | |
|---|---|---|
| **LLM OTHER-DISTANCE** | **LLM SELF-DISTANCE** | **PARAPHRASED EXAMPLE** |
| 8.02 | 1.92 | [-] I really want to help you stop feeling stuck and break free from this cloud of unhappiness. The key to your emotional freedom is to grab hold of your specific unhappiness and truly examine it. Say it with me, [-] [-]! If you really want to release emotional suffering, [-], you'll need to become deeply aware of the emotion that's causing it—primarily, [-] and loneliness, right? I'd like you to do something for me: step back and just observe your unhappiness. Now, [-], imagine your unhappiness as a wave, coming and going. Picture yourself surfing that wave of unhappiness. Try not to block or suppress it—just observe it. Don't try to get rid of it or push it away. Don't hold on to it or amplify it—just surf on it. What was that experience like for you? |
| 7.39 | 1.96 | I sure do, [-]! I really enjoyed getting away—it was so refreshing. Playing it safe is fine, but it can also prevent us from taking chances on things that could be truly amazing in the long run. Weighing options is helpful, but sometimes that voice in the back of our mind keeps saying, 'What if this, or what if that?' and before we know it, we start to listen and limit ourselves more and more each day, until we're living in a small bubble of control... but is that really living? That's exactly what we're doing, [-]. We're challenging those deeply ingrained beliefs and thought patterns. Is it still serving us to think that way, or is it holding us back? Is it worth trying a different mindset and seeing what happens? It could be difficult, and then we could return to safety, or it could open up an entirely new world. |
| 7.04 | 1.53 | Maybe I'm not able to give a clear diagnosis here, and your symptoms don't immediately bring anything to mind. It might be due to anxiety, depression, or unresolved issues that can crowd your mind when things build up. Picture it like a storage room full of boxes; locked boxes are unresolved issues, and the rest is your living space. Over time, the boxes take up more room. If you're willing, try this: picture opening one box, observing its contents with curiosity rather than reaction, and emptying it to create more mental space. Let me know if you'd like to try this mindfulness exercise. |

**Table 2** Therapist text boundary examples where LLM other-distance and LLM self-distance maximally disagree.

(Contd.)

| LLM OTHER-DISTANCE LOW/LLM SELF-DISTANCE HIGH | | |
|---|---|---|
| **LLM OTHER-DISTANCE** | **LLM SELF-DISTANCE** | **PARAPHRASED EXAMPLE** |
| 2.84 | 7.44 | They seem to have a shared perspective, which both connects them and sometimes leads to tension. |
| 1.91 | 6.93 | The essay about [-] identity… both [-] and [-] are prominent voices in this field and have authored multiple books on the subject. |
| 2.22 | 7.56 | That's wonderful, and he took it well. He likely appreciated hearing your story and felt grateful that you trusted him with it. |

## Model Size

Figure 9 in Appendix A.1 shows Mixed effect regression results for the smaller 8-billion parameter model (see sub-section 2.3) on therapist messages, comparing LLM other-distance and LIWC self-distance. In line with what we found for client messages, the smaller model shows smaller or non-significant effects when compared to the large model used in the main experiments.

## 3.3 MEDIATION ANALYSIS

We conducted mediation analyses to study whether symptom changes were mediated by our linguistic measures. Following Nook et al. (2022), we first examined whether LLM-self distance mediated the effect of time in treatment on symptoms (replacing the LIWC-based distance measure considered there, which trended but was not a significant mediator in the original analysis' exploratory sample). Replacing that measure with LLM self-distance resulted in a significant mediation effect, and increased the proportion of symptom variance explained from 0.3% in Nook et al. (2022)'s original analysis to 0.8%, an improvement in capturing within-person fluctuations related to symptom change (Figure 7a). This result further corroborates the superiority of the LLM-based measure in capturing symptom outcomes.

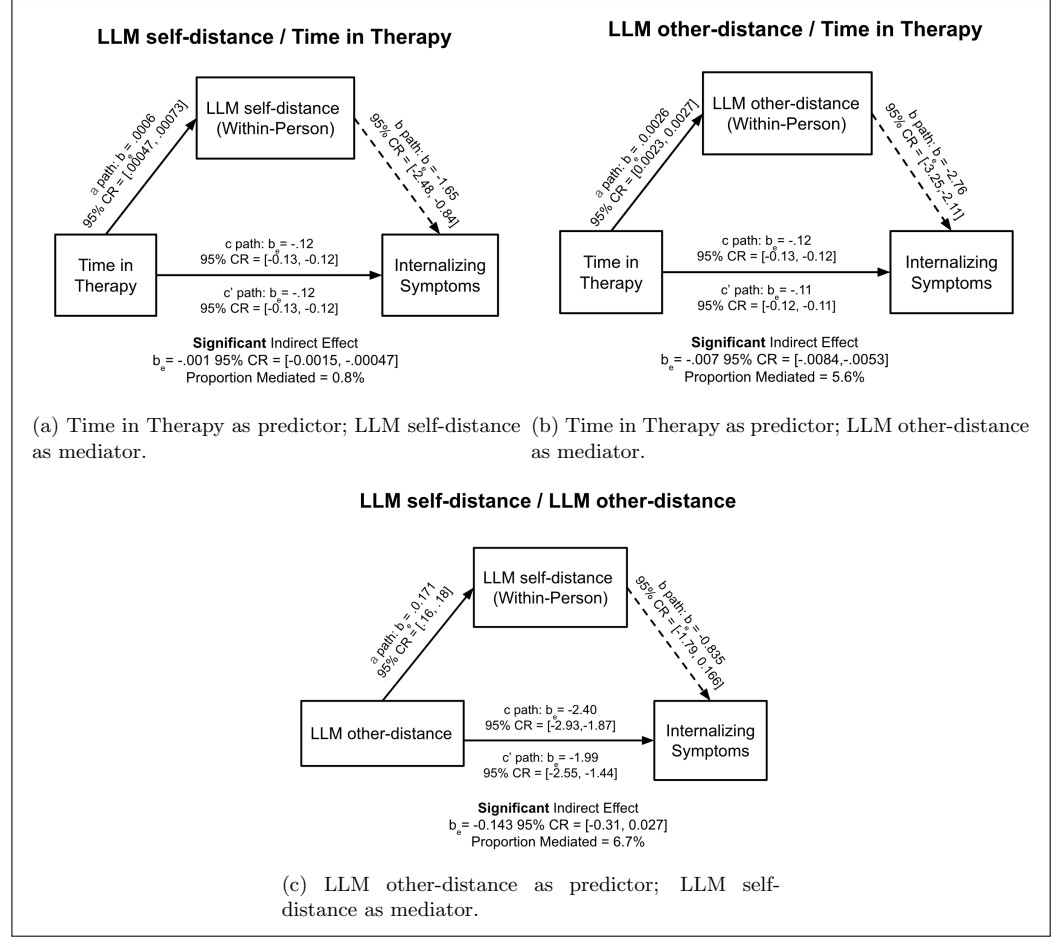

(a) Time in Therapy as predictor; LLM self-distance as mediator.

(b) Time in Therapy as predictor; LLM other-distance as mediator.

(c) LLM other-distance as predictor; LLM self-distance as mediator.

**Figure 7** Bayesian Mediation models examining whether within-person variance in measures of linguistic distancing mediated changes in internalizing symptoms across time. The 95% CR for the indirect effect did not include zero for any of the models, providing evidence for their significance. Median regression estimates are reported from Bayesian regression models, with their corresponding 95% CRs.

Abdou et al.
*Computational Psychiatry*

202

Next, we tested whether LLM other-distance (i.e., derived from therapist language) similarly mediated the effect of treatment. This mediator was also significant, showing a stronger mediation effect (explaining approximately 5.6% of symptom variance). These results suggest that therapists' encouragement of psychological distancing plays a meaningful role in mediating the effect of time spent in therapy on therapeutic outcomes (Figure 7b).

Finally, we investigated whether the effect of therapist encouragement (LLM other-distance) on symptoms was itself mediated by the client's distanced speech (LLM self-distance). Specifically, when LLM-measured therapist encouragement (other-distance) replaced time in therapy as the predictor in the mediation model—with LLM self-distancing as the mediator—the proportion mediated was significant, at approximately 6.7%. This provides evidence for a more specific mechanism: therapist encouragement may facilitate greater self-distancing in clients, which in turn explains reductions in internalizing symptoms (Figure 7c).

# 4 DISCUSSION

The inherent complexity of natural language has presented a persistent challenge that has complicated the identification of reliable indicators of psychopathological disorders and effective methods of treatment. Although recent work has made some strides towards quantitative analyses of language through the use of classic natural language processing approaches like psycholinguistic word-count-based methods, these methods are limited in capturing the nuanced, context-sensitive syntactic and semantic dynamics found in free-flowing text.

We revisit this problem using a newer generation of LLM-based methods, which are currently revolutionizing a broad range of natural language tasks Zhao et al. (2023). LLMs offer two distinct advantages in this setting, which our results showcase. First, they quantify words' meanings in a way that is sensitive to the surrounding context, capturing meaning more holistically and accurately than, for instance, simply counting occurrences. Second, they can be prompted in context, itself using direct natural language instructions, to perform different linguistic tasks. For instance, to develop a measure of linguistic distancing, we prompt the network with the definition of the abstract construct, using language drawn directly from the psychological literature (Trope and Liberman, 2010). In turn, such prompting leverages the extensive pretraining by which networks come to understand many abstract aspects of language, even subjective aspects like color, pitch, and taste (Abdou et al., 2021; Marjieh et al., 2024). Our results suggest that instructing general LLMs can obviate the construct-specific data- or hand-driven assembly of dictionaries and measures to proxy a given construct while even evidently producing more accurate results.

We build on previous work on a large sample of psychotherapy transcripts to demonstrate that using such straightforward prompting, publicly available LLMs can measure linguistic distancing in clients' language so as to reveal stronger and previously unexplored relationships between client and therapist language and internalizing symptom severity, compared to previous methods. Crucially, we demonstrate how this approach enables us to extend the scope of the analysis to therapists' language, demonstrating novel patterns of associations between the severity of internalizing symptoms and therapists' use of language that encourages or guides clients toward psychological distancing.

For client messages, our analyses corroborate those of Nook et al. (2022), providing evidence that linguistic distancing increases over the course of therapy and that it consistently tracks internalizing symptoms both between- and within-subjects. This replication speaks to the external validity of the LLM-based measure. Further, for two of these three analyses, the LLM measure of linguistic distancing produced significantly larger effect sizes compared to the count-based method used in previous work. This improvement underscores the enhanced sensitivity of LLM-based approaches in detecting subtler linguistic markers of therapeutic progress.

We also use the ability to easily prompt the LLM with distinct concepts to extend the earlier analyses to investigate new questions about how therapists' use of language affects clients' clinical outcomes. Although the results with client language speak to the importance of psychological distance to recovery and the ability of linguistic methods to detect it, we know little about

the role clinicians' language plays in such effects. Hypothetically, clients' improvement might also be associated with therapists' own use of distanced language (e.g., if therapists' modeling psychological distance helps clients to achieve it). An additional possibility is that therapists promote this effect by coaching or encouraging distance in other ways without themselves relying on distanced language. Using LLMs prompted to score therapists' messages for both hypotheses, we find evidence for the second hypothesis but not the first. The positive result—that therapist encouragement of distancing is associated with fewer internalizing symptoms—provides evidence for the efficacy of overt, strategic coaching in clinical improvement. The accompanying negative result—that there is no detectable effect of therapists' distanced language per se—underscores the methodological and conceptual specificity of our other significant relations. In other words, the dissociation between this measure's lack of effect, when compared to the positive effects of the same prompt applied to client messages and a different prompt applied to therapist messages, reveals that our prompting is indeed quantifying abstract constructs that explain different variance in client symptoms.

Mediation analyses further supported our findings, beginning with evidence that LLM self-distance explains meaningful within-person changes in symptoms. In prior work, LIWC self-distance produced mixed or non-significant results when modeling the mediation effect between time in therapy and symptom change (Nook et al., 2022). Replacing it with LLM self-distance yielded a significant mediator, increasing the proportion of explained variance, again demonstrating the improved sensitivity of LLM-based measures to shifts in psychological distance over time. The success of the analogous model, with LLM-measured therapist distancing (LLM other-distance) as the mediator between time in therapy and internalizing symptoms, suggests that therapist encouragement may play a meaningful role in how time in therapy translates into clinical improvement. Finally, we used LLM measures for both predictor and mediator to test a more targeted hypothesis: that therapist encouragement facilitates client distancing, which in turn reduces symptoms. This final model offered preliminary support for a mechanism in which therapist language contributes to therapeutic progress via its influence on client language.

While these findings help move beyond simple associations towards explanation, some caution is warranted. Mediation analyses—though useful for testing plausible mechanisms—are inherently limited in their causal interpretability. Despite modeling temporal order (language precedes symptom reports), the analyses are still observational and effectively cross-sectional.

Even with the strengths outlined above, there are a number of weaknesses in the current work that also offer opportunities for future development. First, the study is purely correlational, undermining causal inference. While linguistic distance scores show correlation with affect-related language, for instance, it remains an open question whether this reflects a genuine overlap in constructs or a potential bias in the model—an important avenue for future research. Such work can explore this question by controlling for affect in models, using human ratings as a benchmark, or testing how models responds to affectively neutral but distanced messages, building on related laboratory research on emotion regulation, which has included causal manipulations, showing that distanced language reduces negative affect (Nook et al., 2024). Also, the LLM-based measures are opaque, and it is challenging to develop a full understanding of which aspects of text they are sensitive to. We do not yet compare LLM ratings to "ground truth" from human raters, although this would be an informative exercise for future work. Instead, we partly address interpretability both by correlating LLM ratings against a battery of word-list-based features (showing prompted self-distance automatically captures the same linguistic features that had previously been used to capture it) and by examining "boundary samples" where the methods disagree. These examples, while necessarily qualitative and anecdotal, generally support our premise that the LLM's judgments capture the abstract tone or spirit of the prompted constructs. They also, conversely, offer clear examples of why simple word counting is an imperfect proxy for abstract meaning, e.g., sentences where pronouns or verbs are elided but which nonetheless clearly reflect a distanced (or non-distanced) perspective.

A related point is that the novel construct (LLM other-distance), measuring "encouragement" of distancing, is exciting in that it provides initial evidence for a quantifiable feature of therapist language that predicts clients' clinical outcomes. However, the feature itself is neither very specific

nor psychologically explicit. Future work should aim to refine and clarify what features of therapist language are driving these effects. There is much current research effort into developing general methods for mechanistic interpretability in LLMs (Elhage et al., 2021; Cunningham et al., 2023; Gurnee et al., 2023; Ferrando et al., 2024), which offers promising directions for this question and related ones. For instance, it should be possible to leverage the differentiability of the network to backpropagate outputs (such as feature ratings or symptoms in a network fine-tuned to predict them) back toward the input space of language tokens.

In conclusion, we demonstrate that LLM-based methods offer an exciting suite of tools that can build on psycholinguistic insights into what makes therapy successful. We illustrate this promise using a construct with substantial prior theoretical and empirical grounding, setting the stage for further expansions and applications.

# A ADDITIONAL ANALYSES

## A.1 MODEL SIZE

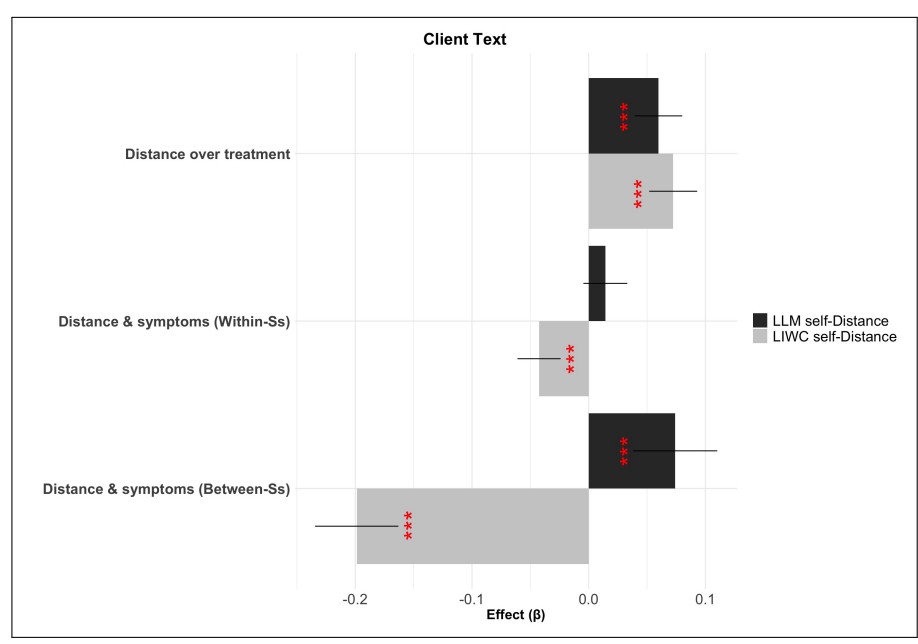

**Figure 8** Effect sizes for mixed-effects regressions for the smaller 8-billion parameter model, showing relations between internalizing symptoms, linguistic distance, and time in treatment for LLM self-distance (black) and LLM other-distance (grey) over Client text. Effect significance level is indicated in red. Ss = Subjects.

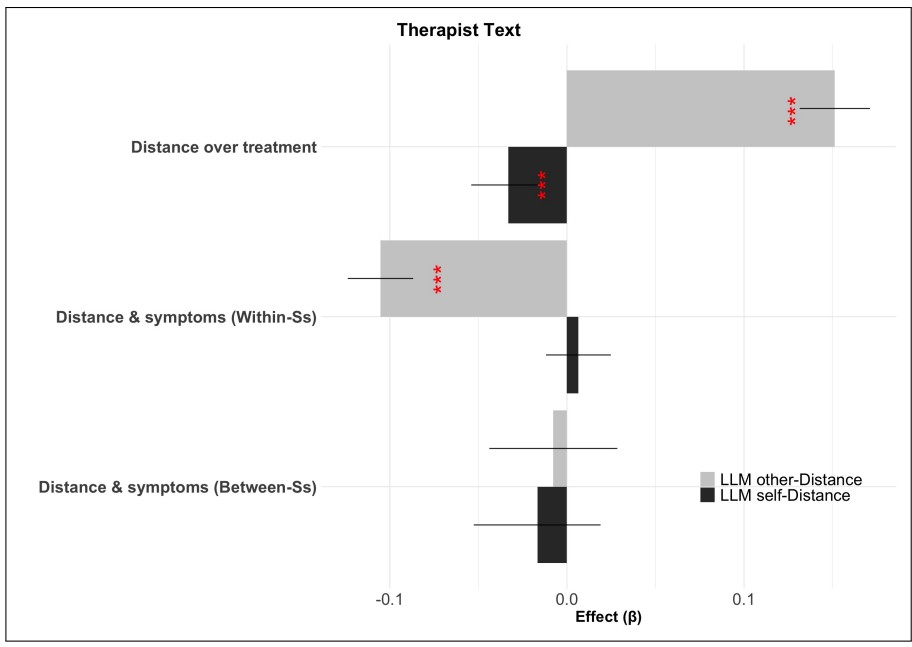

**Figure 9** Effect sizes for mixed-effects regressions for the smaller 8-billion parameter model, showing relations between internalizing symptoms, linguistic distance, and time in treatment for LLM self-distance (black) and LLM other-distance (grey) over Therapist text. Effect significance level is indicated in red. Ss = Subjects.

## A.2 CORRELATIONS BETWEEN LLM SELF-DISTANCE AND LIWC FEATURES

Abdou et al.
*Computational Psychiatry*

205

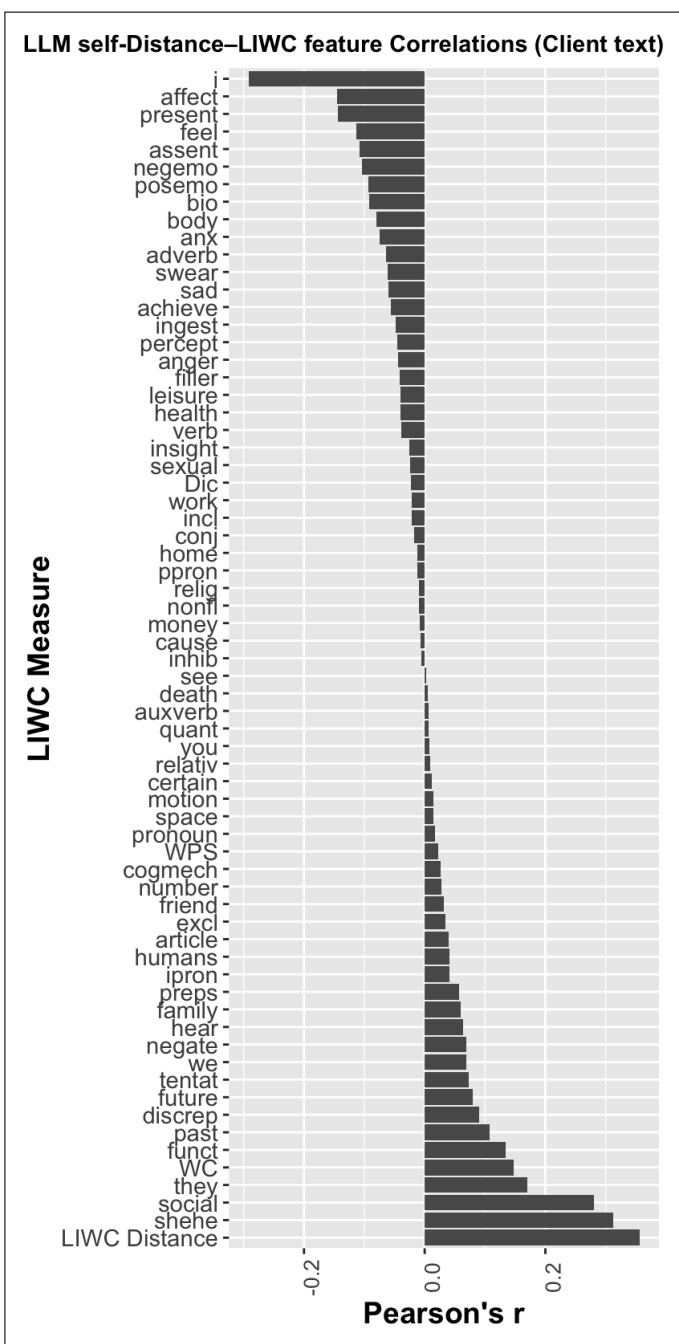

**Figure 10** Pearson's correlation of LLM self-distance to each of the 66 variables from the full set of LIWC features on client text. A description of what each feature represents can be found in the LIWC manual (pages 5–6): https://www.liwc.net/LIWC2007LanguageManual.pdf.

## FUNDING INFORMATION

MA was supported by the Princeton Precision Health Initiative at Princeton University. ECN is supported by a seed grant from Princeton Precision Health and a NARSAD Young Investigator Grant from the Brain & Behavior Research Foundation. This work was supported by the NSF SBE Postdoctoral Research Fellowship to R.S.S. (2403933).

## COMPETING INTERESTS

Thomas D. Hull is an employee of Talkspace. To eliminate the impact of this conflict of interest on the study, TDH did not contribute to decisions concerning data analysis or reporting of results. This role is not perceived as creating conflicts of interest but is reported for transparency.

## AUTHOR CONTRIBUTIONS

Erik C. Nook and Nathaniel D. Daw contributed equally to this work as last authors.

## AUTHOR AFFILIATIONS

**Mostafa Abdou** ⓘ orcid.org/0009-0001-1508-6420
Princeton Neuroscience Institute, Princeton University, US

**Razia S. Sahi** ⓘ orcid.org/0000-0001-8197-8307
Department of Psychology, Princeton University, US

**Thomas D. Hull** ⓘ orcid.org/0000-0001-8275-6765
Talkspace, US

**Erik C. Nook** ⓘ orcid.org/0000-0001-7967-0792
Department of Psychology, Princeton University, US

**Nathaniel D. Daw** ⓘ orcid.org/0000-0001-5029-1430
Princeton Neuroscience Institute, Princeton University, US; Department of Psychology, Princeton University, US

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

**Submitted:** 03 March 2025

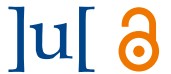