## [Reviewer Report · Peer Review History]

## Comments/Explanation (required). Please include references to any minor or major revisions required, as well as addressing any issues with language clarity. Authors will see these comments:

This paper uses a large language model to examine transcripts from psychotherapy. The aims are two-fold. First, the authors aim to compare an LLM analysis, whereby the LLM is ‘prompted’ in the form of language to a more transparent and classical analysis using word counts. Second, their aim is to validate the specificity of the LLM findings. The methods are entirely appropriate, using a large dataset of mostly text-based psychotherapy interactions, and a recent open-source large language model. They find, first, that LLM analyses lead to slightly better predictions of clinical outcomes than more standard word count-based analyses. Second, they extend their analyses to examine not only clients in the therapy, but also the therapist language. Critically, they are able to distinguish how therapists themselve use distancing, to how they encourage distancing; and are able to demonstrate distinct relationships to clinical outcomes.

This paper was a real pleasure to read. I am afraid I have no meaningful concerns to raise. The authors have a clear set of relevant questions, and approach them with precision and clarity. The writing and results are as clear as they could be, and the discussion entirely appropriate and enlightening. I have some thoughts, but these are really just food for thoughts that the authors may or may not want to address. I think the paper is essentially fine as it is.

The main point I was left unsure about is the process of prompt engineering. This is itself a highly complex thing in that it effectively forms part of the measurement, being part of the input to the network. Examining this is also very costly as it requires sampling from the model. The distinction between the self and other findings on the therapist data hinges on differences in the prompts. On the one hand, the work goes as far as I think we can really go at present, in terms of validating the output, and this is great. However, the reliance on the many degrees of freedom in the prompt does also raise the question of whether some of the subtler relationships might depend on the details of the prompt. Ultimately, it is likely that say an adversarial prompt of some sort likely exists which would yield very different results, but I am not sure if that matters.

The other point I was left wondering was whether it might be possible to go just a little further towards mechanisms and causality by attempting a mediation analysis of sorts, asking whether the clinical improvement due to encouragement is mediated by change in distancing observed over sessions in the clients. From reading the paper, I think this analysis could be doable.

The final question or thought I had was whether, given the sample size, distinct relationships to core symptoms of mood vs anxiety could be examined. So far, the analyses focus on summed PHQ-8 and GAD-7 scores. However, treatment for depression and anxiety does differ, and the question of how effective distancing is in both domains is clinically intersting. Do the effects, for instance, look the same whether one were to focus on subgroups with predominantly high PHQ vs GAD scores?

## Peer-review-recommendation

Accept Submission

---

## [Reviewer Report · Peer Review History]

## Recommendation:

Accept with Minor Revision

## Comments/Explanation (required). Please include references to any minor or major revisions required, as well as addressing any issues with language clarity. Authors will see these comments:

General comments:

The paper tackles a very relevant and useful question using a novel and innovative methodologly. The statistical analyses conducted are diligent and very well documented. Overall, the article is very well written. The methods are easy to follow and the flow of the article is very compelling. Particularly the description of the LLM they used is very clear detailed. Some of the figures could use some work as well as some minor discussion points (see detailed comments below). Also, the figure titles could be a bit more informative to make the figures more easily understood at first glance.

1. Introduction: The authors cite a range of studies that applied LLMs to psychiatric research, but do not reference the content of these studies. It would be interesting to add a sentence or two on examplary questions that were asked using LLMs in psychiatry research.

2. Introduction or discussion: When discussion the link between cognitive distancing and internalising symptoms, Dercon (2024) would be a relevant source to cite. This study shows effects of an online distancing intervention.

3. Methods, Dataset: You say that the clients were randomly sampled. Were there no restrictions applied to this, such as minimum length of conversations, minimum frequency of messages, minimum completion of questionnaires? The number of 2, 875 clients seems rather arbitrary, thus it would be good to clarify where this number came from.

4. Methods, Symptom assessment: The authors use a sum score of the PHQ-8 and the GAD-7. This seems like a reasonable und not uncommon approach. Nevertheless, it would be helpful to report the strength of the correlation between the two measures (it just says it’s high) and maybe to cite papers that showed that depression and anxiety tend to cluster together when performing factor analysis (e.g. gillan et al. 2016, Wise et al. 2023). 

5. Methods, linguistic distancing: The authors mention that linguistic distancing was calculated for each message separately and then averaged across messages. It seems slightly odd that all messages received the same weight even though some messages might have been considerably shorter than others. It would be helpful if the authors could comment on this and whether they think there are considerable and important differences in the length of the messages.

5.1 For the LIWC distance, it is not entirely clear whether the counts were performed entirely manually or whether at least parts were done through regular expressions or similar automatisations.

6. Methods, Model prompting: The authors mention that the prompts to the LLM were “truncated after 1000 tokens due to empirically 213 determined GPU memory limits". It is somewhat unclear what is meant here. Was the text for the model to evaluate truncated or was the entire prompt truncated or just the rating instructions? For how many of the messages was this the case? I think this could be problematic as the model could be missing some information. Further, excessive context can strongly impair generative model performance. At the same time, the instruction prompts as described in the article see very long and excessively vague. The final extracted ratings do suggest that this prompting technique was working, but I cannot help but wonder whether performance could be improved by stating the LLM’s task more concisely, possibly by providing a list of aspects to look out for, similar to the count-based approach. While the ratings seem reliable enough that I do not deem it necessary for the authors to repeat their analyses using a more concise prompt that would not require any truncation, I do believe that it would be important to clarify a) what exactly was truncated, b) In how many cases this had to be done, c) What influenced the choice of instruction prompt.

7. Methods, statistical analyses: It is not entirely clear what the authors mean by a decomposition of between-subject and within-subject relations of distancing and internalising traits. It would be helpful to add a short explanation of this procedure, beyond referring to Nook et al.

8. Figure 3: This figure is referenced as Fig. 3 top, but there is only one part to the figure. It would however be a nice idea to combine it with figure 4 into one larger figure. More importantly, the usage of the ggplot default theme is not particularly aesthetically pleasing to me, especially because it is inconsistent with Figure 4 which seems to be using theme_minimal(). Further, the scatter points are too many to be actually informative. The figure would be improved by removing the black scatter points and only keeping the orange points and the line. The caption should however include an explanation of what the orange points stand for. They seem to be some kind of average but it is not entirely clear.

9. Figure 8: This figure is referenced right after Figure 4 but can only be found in the end of the paper. If the figure is meant to be supplemental, it should be added to the appendix and referenced as supplemental figure. If it is meant to be a main text figure, it should be placed closer to where it is mentioned in the text. The same is true for Figure 9. Further, the figure is currently hard to understand on its own, without the full text. The labels on the y axis are confusing, so it might be helpful to bin related words/concepts. Further, as mentioned in the general comments, this figure in particular but all figures in general could use a more informative title to make their message more apparent at first glance.

10. Results, correlations with other measures: The authors mention that the LLM-based distancing scores were also correlated with affect-related words. This speaks towards a potential bias in the model to label more positively-valenced text as more distanced and vice-versa. This also seems to be the case in the boundary examples. It would be beneficial for the authors to discuss this potential shortcoming and discuss ways to investigate this possible bias.

11. Results, model size: Here, you show a plot that replicates the regression results using the 8B version of the LLM. While this is an interesting and relevant result, it would be better suited in the appendix. For the main text, it would be helpful to plot the correlation between the distancing measure of the 8B model and the count-based measure, similar to figure 3, to show how well the 8B model can capture the distancing measure. This is also true for the later section on the model size related to the therapist’s messages.

12. Figure 6 has the same problems as figure 3.

13. Figure 9: The title of the figure or at least the caption should mention that this pertains to the smaller model. On its own it is otherwise a bit confusing.

In conclusion, I strongly recommend acceptance of this paper once these minor comments have been addressed. It contains very diligent work that represents an important contribution to the field of LLM applications to psychiatric research and to the field of computational psychiatry more broadly.

---

## [Reviewer Report · Peer Review History Round 2, Quentin Huys]

## Reviewer F

Prof. Mona Garvert

Associate Editor

Computational Psychiatry

Dear Prof. Garvert

Thank you very much for your invitation to revise and resubmit our paper entitled “Leveraging large language models to estimate clinically relevant psychological constructs in psychotherapy transcripts.” We sincerely appreciated both the positive reviews and constructive suggestions we received from reviewers. We have incorporated these edits and believe they have strengthened the manuscript considerably. Attached you will find a detailed response to all comments and a revised version of the manuscript. Changes are noted in red text. We hope you find this revised paper suitable for publication in Computational Psychiatry. Please do not hesitate to contact me with any questions.

Sincerely,

Mostafa Abdou

Postdoctoral Researcher

Princeton Neuroscience Institute

Princeton University

1. This paper uses a large language model to examine transcripts from psychotherapy. The aims are two-fold. First, the authors aim to compare an LLM analysis, whereby the LLM is ‘prompted’ in the form of language to a more transparent and classical analysis using word counts. Second, their aim is to validate the specificity of the LLM findings. The methods are entirely appropriate, using a large dataset of mostly text-based psychotherapy interactions, and a recent open-source large language model. They find, first, that LLM analyses lead to slightly better predictions of clinical outcomes than more standard word count-based analyses. Second, they extend their analyses to examine not only clients in the therapy, but also the therapist language. Critically, they are able to distinguish how therapists themselves use distancing, to how they encourage distancing; and are able to demonstrate distinct relationships to clinical outcomes. This paper was a real pleasure to read. I am afraid I have no meaningful concerns to raise. The authors have a clear set of relevant questions, and approach them with precision and clarity. The writing and results are as clear as they could be, and the discussion entirely appropriate and enlightening. I have some thoughts, but these are really just food for thoughts that the authors may or may not want to address. I think the paper is essentially fine as it is.

## Author’s response

We are grateful for the reviewers’ kind words and thoughtful suggestions.

## Reviewer F

2. The main point I was left unsure about is the process of prompt engineering. This is itself a highly complex thing in that it effectively forms part of the measurement, being part of the input to the network. Examining this is also very costly as it requires sampling from the model. The distinction between the self and other findings on the therapist data hinges on differences in the prompts. On the one hand, the work goes as far as I think we can really go at present, in terms of validating the output, and this is great. However, the reliance on the many degrees of freedom in the prompt does also raise the question of whether some of the subtler relationships might depend on the details of the prompt. Ultimately, it is likely that say an adversarial prompt of some sort likely exists which would yield very different results, but I am not sure if that matters.

## Author’s response

We agree this is an important point. Prompt design introduces meaningful degrees of freedom that influence model outputs, and even small changes in phrasing can have non-trivial effects. We chose an approach informed by literature on prompt design (as is now noted on page 6, lines 217-220), but we fully agree that systematic exploration of prompt sensitivity would make for valuable future work.

“We used a detailed prompt to prioritize conceptual clarity and ensure task alignment. While excessively long prompts can impair model performance, prompt-engineering research shows that well-structured, moderately long prompts often enhance LLM performance—particularly for complex, domain-specific tasks (Liu et al., 2025; Long et al., 2025).”

## Reviewer F

3. The other point I was left wondering was whether it might be possible to go just a little further towards mechanisms and causality by attempting a mediation analysis of sorts, asking whether the clinical improvement due to encouragement is mediated by change in distancing observed over sessions in the clients. From reading the paper, I think this analysis could be doable.

## Author’s response

This is a valuable suggestion that we believe has strengthened the manuscript. We followed the reviewer’s recommendation and conducted mediation analyses. The original Nook et al. (2022) paper reported a mediation model of LIWC client self-distancing scores showing that it did not significantly mediate symptom changes over time. We first replicated this model, substituting our LLM self-distance measure in place of LIWC self-distance.

We then extended the mediation analysis to examine the role of therapist language. In a second model, we replaced the mediator (LLM self-distance) with therapist LLM other-distance. This mediation model was also significant, and interestingly, the effect was substantially larger, with approximately 5.6% of symptom variance explained via therapist encouragement. Finally, we tested a model in which LLM other-distance replaced time in therapy as the predictor. This model asks whether there is evidence to support a model in which therapist encouragement increases client linguistic distance, which then results in lower symptoms. This model was also significant, and the proportion of variance mediated was 6.7%, suggesting that therapist encouragement of distancing may serve as an interpersonal mechanism of change.

For method description see page 11, lines 312-336; for results see page 20, lines 490-503 for discussion see 562-578.

**Figure 7 d67e5080:** Bayesian Mediation models examining whether within-person variance in measures of linguistic distancing mediated changes in internalizing symptoms across time. The 95% CR for the indirect effect did not include zero for any of the models, providing evidence for their significance. Median regression estimates are reported from Bayesian regression models, with their corresponding 95% CRs.

## Reviewer F

4. The final question or thought I had was whether, given the sample size, distinct relationships to core symptoms of mood vs anxiety could be examined. So far, the analyses focus on summed PHQ-8 and GAD-7 scores. However, treatment for depression and anxiety does differ, and the question of how effective distancing is in both domains is clinically interesting. Do the effects, for instance, look the same whether one were to focus on subgroups with predominantly high PHQ vs GAD scores?

## Author’s response

We agree this is an important and clinically relevant question. There were several reasons why we chose to focus on overall internalizing symptoms. First, these are highly correlated scores (*r* = 0.757) and thus are likely tracking a single dimension of internalizing symptoms in this dataset. Second, in the original report (Nook et al. 2021), supplemental analyses investigated PHQ-8 and GAD-7 separately and found almost no differences between them. Third, we wished to keep the scope of analyses focused; analyzing these scores separately would triple the number of tests to report. Given these three points, we worried that analyzing them separately would add a lot of bloat to the paper without adding new information. We believe pursuing these finer-grained distinctions is outside the scope of the current paper.

---

## [Reviewer Report · Peer Review History Round 2, Kristin Witte]

## Reviewer I

0. The paper tackles a very relevant and useful question using a novel and innovative methodologly. The statistical analyses conducted are diligent and very well documented. Overall, the article is very well written. The methods are easy to follow and the flow of the article is very compelling. Particularly the description of the LLM they used is very clear detailed. Some of the figures could use some work as well as some minor discussion points (see detailed comments below). Also, the figure titles could be a bit more informative to make the figures more easily understood at first glance.

## Author’s response

We thank the reviewer for their encouraging and helpful feedback. We’re pleased that the methodological clarity and flow of the paper came through. We have revised some of the figures to make them more informative and consistent, and we have addressed specific issues in the point-by-point responses that follow.

## Reviewer I

1. Introduction: The authors cite a range of studies that applied LLMs to psychiatric research, but do not reference the content of these studies. It would be interesting to add a sentence or two on examplary questions that were asked using LLMs in psychiatry research.

## Author’s response

We added a brief overview of the content of some of this research on page 2, lines 68-70.

“Here, we contribute to an emerging body of research that leverages these methods for psychological and psychiatric research—for example, to detect suicidal ideation in social media posts, generate interpretable symptom-level predictions from online text, and identify symptom-relevant content and summaries from clinical interviews.”

## Reviewer I

2. Introduction or discussion: When discussion the link between cognitive distancing and internalising symptoms, Dercon (2024) would be a relevant source to cite. This study shows effects of an online distancing intervention.

## Author’s response

We added a citation (page3, line 81): “these distancing techniques are thought to help individuals adopt a perspective with a higher level of construal (Trope and Liberman, 2010; Moran and Eyal, 2022; Dercon et al., 2024)”

## Reviewer I

3. Methods, Dataset: You say that the clients were randomly sampled. Were there no restrictions applied to this, such as minimum length of conversations, minimum frequency of messages, minimum completion of questionnaires? The number of 2,875 clients seems rather arbitrary, thus it would be good to clarify where this number came from.

## Author’s response

We corrected a discrepancy in the manuscript to reflect that our analysis included 3,727 clients, and that we followed the selection reported in Nook et al. (2022). We also added a brief explanation of these criteria and clarified that our analyses were limited to the exploratory dataset used in the original study (page 4, lines 134-137):

“Following Nook et al. (2022), participants were included only if they completed at least three symptom assessments over a minimum span of six weeks. After applying these selection criteria, Nook et al. (2022) divided the remaining pool of clients (6, 229) into exploratory (3, 727) and validation sets (2,500). Only the former is utilized in this study”

## Reviewer I

4. Methods, Symptom assessment: The authors use a sum score of the PHQ-8 and the GAD-7. This seems like a reasonable und not uncommon approach. Nevertheless, it would be helpful to report the strength of the correlation between the two measures (it just says it’s high) and maybe to cite papers that showed that depression and anxiety tend to cluster together when performing factor analysis (e.g. gillan et al. 2016, Wise et al. 2023).

## Author’s response

We now note the strong correlation observed in our dataset between the two (*r* = 0.757) and cite prior work using dimensional approaches in psychopathology research to justify the composite score (page 5, lines 169-173):

“Following Nook et al. (2022), we summed the two measures to produce a single assessment of internalizing symptoms, as PHQ-8 and GAD-7 measures were highly correlated (r = 0.757, p < 0.001). This strong relationship aligns with prior work showing that depression and anxiety commonly co-occur and load onto a shared internalizing factor in dimensional models of psychopathology (Gillan et al., 2016; Newman, 2022; Wise et al., 2023).”

## Reviewer I

5. Methods, linguistic distancing: The authors mention that linguistic distancing was calculated for each message separately and then averaged across messages. It seems slightly odd that all messages received the same weight even though some messages might have been considerably shorter than others. It would be helpful if the authors could comment on this and whether they think there are considerable and important differences in the length of the messages.

## Author’s response

This is a thoughtful point. We now clarify that we followed Nook et al. (2022) in averaging message-level scores without weighting by length for the sake of comparability and to avoid overemphasizing longer messages. We also added a note that message length (measured as word count) is modestly correlated with LLM self-distance (r = .16, p < 0.001), but the significance of our results does not change when controlling for it (page 6, lines 213-218.

“Distancing scores were averaged across messages without weighting by length, following Nook et al. (2022). While this gives equal weight to short and long messages, we retained their approach for comparability and to avoid overemphasizing longer messages that may not be more informative. Nevertheless, we note that message length (measured as a word count) does show some correlation with LLM self-distance (r = 0.16; p < .001), but the significance of our results does not change when controlling for word count.”

## Reviewer I

6.1 For the LIWC distance, it is not entirely clear whether the counts were performed entirely manually or whether at least parts were done through regular expressions or similar automatisations.

## Author’s response

We now clarify in the Methods that all LIWC scores were generated automatically using the LIWC software, which applies dictionary-based word counts programmatically. No manual annotation or regular expressions were used. (page 6, line 196):

“Verb and pronoun counts were computed programmatically using LIWC software.”

## Reviewer I

7. Methods, Model prompting: The authors mention that the prompts to the LLM were “truncated after 1000 tokens due to empirically determined GPU memory limits”. It is somewhat unclear what is meant here. Was the text for the model to evaluate truncated or was the entire prompt truncated or just the rating instructions? For how many of the messages was this the case? I think this could be problematic as the model could be missing some information. Further, excessive context can strongly impair generative model performance. At the same time, the instruction prompts as described in the article see very long and excessively vague. The final extracted ratings do suggest that this prompting technique was working, but I cannot help but wonder whether performance could be improved by stating the LLM’s task more concisely, possibly by providing a list of aspects to look out for, similar to the count-based approach. While the ratings seem reliable enough that I do not deem it necessary for the authors to repeat their analyses using a more concise prompt that would not require any truncation, I do believe that it would be important to clarify a) what exactly was truncated, b) In how many cases this had to be done, c) What influenced the choice of instruction prompt.

## Author’s response

We agree that these are important points to clarify. We added a footnote (page 7, line 236) explaining: “Only the message text was truncated after 1,000 tokens, meaning that all text that is part of a message that exceeded the first 1,000 tokens was discarded. The instruction prompt before and after the message was kept fully intact. Such truncation occurred in only 0.55% of messages”.

Regarding prompt design, we added a few lines to the manuscript explaining that our goal was to ensure clarity and alignment with the intended construct, prioritizing task fit over brevity (page 7, lines 220-224). As we now note, recent research suggests that well-structured, moderately long prompts can enhance LLM performance on complex, domain-specific tasks (even though excessively long prompts can, indeed, hurt performance).

“We used a detailed prompt to prioritize conceptual clarity and ensure task alignment. While excessively long prompts can impair model performance, prompt-engineering research shows that well-structured, moderately long prompts often enhance LLM performance—particularly for complex, domain-specific tasks (Liu et al., 2025; Long et al., 2025)”

## Reviewer I

8. Methods, statistical analyses: It is not entirely clear what the authors mean by a decomposition of between-subject and within-subject relations of distancing and internalising traits. It would be helpful to add a short explanation of this procedure, beyond referring to Nook et al.

## Author’s response

We added a sentence clarifying how between-subject and within-subject are defined. (page 10, lines 265-271): “Between-person effects were defined as a participant’s average linguistic distance grand mean centered (i.e., their average deviation from the group), while within-person effects are observation scores after participant-centering (i.e., differences between the observation and a participant’s mean). When entered together in a regression, these variables allowed us to test whether variance in the outcome variable was associated with person-level variance relative to other clients (between-subjects component) and/or observation-level variance relative to the client’s own mean (within-person component).”

## Reviewer I

9. Figure 3: This figure is referenced as Fig. 3 top, but there is only one part to the figure. It would however be a nice idea to combine it with figure 4 into one larger figure. More importantly, the usage of the ggplot default theme is not particularly aesthetically pleasing to me, especially because it is inconsistent with Figure 4 which seems to be using theme_minimal(). Further, the scatter points are too many to be actually informative. The figure would be improved by removing the black scatter points and only keeping the orange points and the line. The caption should however include an explanation of what the orange points stand for. They seem to be some kind of average but it is not entirely clear.

## Author’s response

Thank you for these helpful suggestions. We have updated Figure 3 to use a consistent visual theme (theme_minimal) and removed the dense black scatter points to improve clarity. We also revised the figure caption to explain that the orange points represent bin-averaged values with corresponding regression lines. We also corrected the in-text reference to match the revised figure format.

**Figure 3 d67e5189:** Correlation between LLM self-distance and LIWC self-distance on 100,000 randomly sampled examples from the dataset. Orange points represent the mean LIWC self-distance within one of 100 bins of the LLM self-distance measure. A linear regression line is overlaid in blue.

## Reviewer I

9. Figure 8: This figure is referenced right after Figure 4 but can only be found in the end of the paper. If the figure is meant to be supplemental, it should be added to the appendix and referenced as supplemental figure. If it is meant to be a main text figure, it should be placed closer to where it is mentioned in the text. The same is true for Figure 9. Further, the figure is currently hard to understand on its own, without the full text. The labels on the y axis are confusing, so it might be helpful to bin related words/concepts. Further, as mentioned in the general comments, this figure in particular but all figures in general could use a more informative title to make their message more apparent at first glance.

## Author’s response

We have moved both figures to the appendix, and we have added additional information to the caption (for figure 9).

“Figure 9: Pearson’s correlation of LLM self-distance to each of the 66 variables from the full set of LIWC features on client text. A description of what each feature represents can be found in the LIWC manual (pages 5-6): https://www.liwc.net/LIWC2007LanguageManual.pdf.”

## Reviewer I

10. Results, correlations with other measures: The authors mention that the LLM-based distancing scores were also correlated with affect-related words. This speaks towards a potential bias in the model to label more positively-valenced text as more distanced and vice-versa. This also seems to be the case in the boundary examples. It would be beneficial for the authors to discuss this potential shortcoming and discuss ways to investigate this possible bias.

## Author’s response

This is an astute observation that we believe would be best addressed in papers beyond the current report here. Our paper focuses on distancing-symptoms relationships and believe the relationship between distancing and affect is an important question for future research, particularly in exploring process-level mechanisms. We have added a line in the Discussion noting that while LLM-based distancing scores correlate with affect-related language, it remains unclear whether this reflects a true conceptual overlap or a modeling bias. We now explicitly mention this as a direction for future work, where the relationship can be explored “by controlling for affect in models, using human ratings as a benchmark, or testing how the model responds to affectively neutral but distanced messages.” (page 23, lines 583-586)

## Reviewer I

11. Results, model size: Here, you show a plot that replicates the regression results using the 8B version of the LLM. While this is an interesting and relevant result, it would be better suited in the appendix. For the main text, it would be helpful to plot the correlation between the distancing measure of the 8B model and the count-based measure, similar to figure 3, to show how well the 8B model can capture the distancing measure. This is also true for the later section on the model size related to the therapist’s messages.

## Author’s response

We have moved both plots to the appendix.

## Reviewer I

12. Figure 6 has the same problems as figure 3.

## Author’s response

We have implemented the same changes for figure 6 (now figure 5) as for figure 3.

**Figure 5 d67e5221:** Correlation between LLM other-distance and LLM self-distance on 100,000 randomly sampled examples from the dataset. Orange points represent the mean LIWC self-distance within one of 100 bins of the LLM self-distance measure. A linear regression line is overlaid in blue.

## Reviewer I

13. Figure 9: The title of the figure or at least the caption should mention that this pertains to the smaller model. On its own it is otherwise a bit confusing.

## Author’s response

We have implemented this change (to what is now, after reorganizing, figure 8). The caption now reads:

“Figure 8: Effect sizes for mixed-effects regressions for the smaller 8-billion parameter model, showing relations between internalizing symptoms, linguistic distance, and time in treatment for LLM self-distance (black) and LLM other-distance (grey) over Therapist text. Effect significance level is indicated in red. Ss = Subjects.”

## Reviewer I

14. In conclusion, I strongly recommend acceptance of this paper once these minor comments have been addressed. It contains very diligent work that represents an important contribution to the field of LLM applications to psychiatric research and to the field of computational psychiatry more broadly.

## Author’s response

We are grateful for the reviewer’s supportive and constructive comments on our paper.